# Reversal of contractility as a signature of self-organization in cytoskeletal bundles

Martin Lenz[1,2]*

[1]Université Paris-Saclay, CNRS, LPTMS, Orsay, France; [2]PMMH, CNRS, ESPCI Paris, PSL University, Sorbonne Université, Université de Paris, Paris, France

**Abstract** Bundles of cytoskeletal filaments and molecular motors generate motion in living cells, and have internal structures ranging from very organized to apparently disordered. The mechanisms powering the disordered structures are debated, and existing models predominantly predict that they are contractile. We reexamine this prediction through a theoretical treatment of the interplay between three well-characterized internal dynamical processes in cytoskeletal bundles: filament assembly and disassembly, the attachement-detachment dynamics of motors and that of crosslinking proteins. The resulting self-organization is easily understood in terms of motor and crosslink localization, and allows for an extensive control of the active bundle mechanics, including reversals of the filaments' apparent velocities and the possibility of generating extension instead of contraction. This reversal mirrors some recent experimental observations, and provides a robust criterion to experimentally elucidate the underpinnings of both actomyosin activity and the dynamics of microtubule/motor assemblies in vitro as well as in diverse intracellular structures ranging from contractile bundles to the mitotic spindle.

*For correspondence:
martin.lenz@u-psud.fr

**Competing interests:** The author declares that no competing interests exist.

## Introduction

Many cellular functions, from motility to cell division, are driven by molecular motors exerting forces on actin filaments or microtubules held together by crosslinking proteins. This wide variety of processes is powered by an equally wide range of structures, many of which do not display any apparent spatial organization of their components (*Verkhovsky et al., 1995*; *Cramer et al., 1997*; *Medalia et al., 2002*; *Kamasaki et al., 2007*). While actomyosin structures are overwhelmingly observed to contract (*Murrell et al., 2015*), the mechanisms underlying this contraction are unclear, as individual motors can in principle elicit extension just as easily as contraction (*Figure 1a–b*; *Hatano, 1994*; *Sekimoto and Nakazawa, 1998*; *Lenz et al., 2012a*; *Mendes Pinto et al., 2013*).

Recent investigations into this breaking of symmetry between contraction and extension have focused on three classes of models. The first one is specific to actin filaments, which are very flexible, as it is based on the idea that *mechanical nonlinearities*, for example, the buckling of individual filaments under compression could suppress the propagation of extensile forces and thus favor contraction (*Dasanayake et al., 2011*; *Lenz et al., 2012b*; *Ronceray et al., 2016*). The second mechanism is relevant for microtubule-based systems, where some (but not all) motors may *dwell on the ends of the filaments*, which transiently generating a type of organization similar to that found in muscle (*Foster et al., 2015*; *Tan et al., 2018*). Finally, in the third type of models the spatial *self-organization* of the bundle's motors and crosslinks along undeformable, rod-like actin filaments leads to contraction (*Kruse and Sekimoto, 2002*; *Zumdieck et al., 2007*; *Zemel and Mogilner, 2009*; *Oelz et al., 2015*; *Koenderink and Paluch, 2018*). So far, opportunities to discriminate between these models experimentally remain very limited for lack of a clear theoretical prediction setting one apart from the others.

Here we provide such a prediction, namely that the self-organization mechanism implies that filament-motor bundles robustly *extend* if taken to certain parameter regimes. This stark qualitative

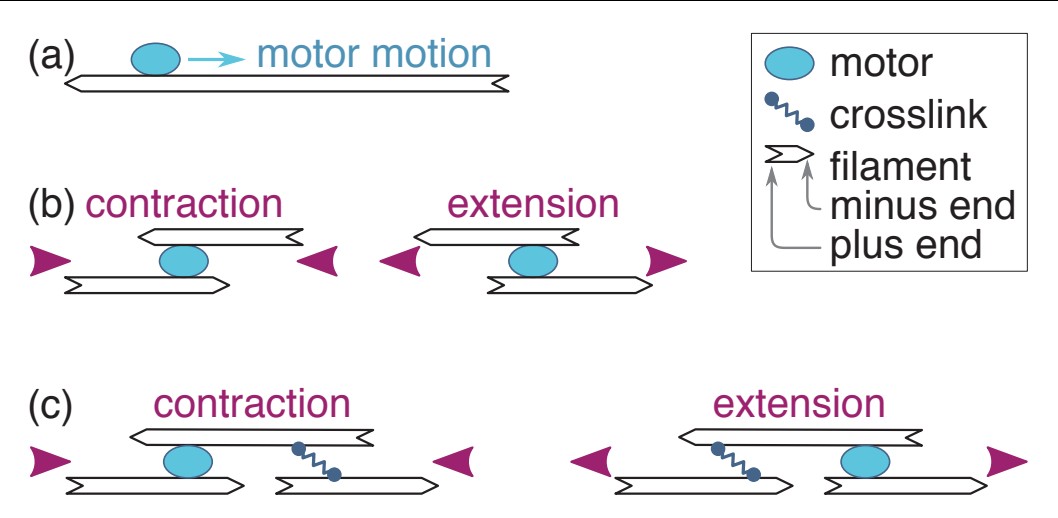

**Figure 1.** Active motor-filament bundle dynamics involves a competition between contraction and extension. (**a**) Motors bind filaments and move towards their plus ends. (**b**) This motion results in local contraction or extension depending on the local arrangement of the filaments. (**c**) In a full bundle, a given filament arrangement can generate contraction or extension depending on the localization of the motors and crosslinks. The present work shows that motor and crosslink self-organization can bring about either outcome.

change from contraction to extension is easily detectable experimentally and is not expected in models based on mechanical nonlinearities or motor dwelling. This prediction could also explain the ill-understood extension observed in some microtubule systems (*Sanchez et al., 2012*; *Keber et al., 2014*; *Roostalu et al., 2018*). Our prediction crucially rests on a simultaneous theoretical treatment of the filament polymerization, motor and crosslink dynamics detailed in Sec. 'Model'. Previous studies of contractility mechanisms only involved partial treatments, whereby the time scale associated with one of these dynamics was effectively assumed to be much larger or smaller than the others. We show in Sec. 'Self-organization and force distribution' that the coupled dynamics of these three elements induces a spatial organization of motors and crosslinks along the filaments. Sec. 'Velocity selection and bundle tension' then demonstrates that the resulting localization of the motors and crosslinks in the vicinity of the filament ends leads to a switch between contraction and extension, as schematized in *Figure 1c*. In Sec. 'Qualitative predictions' , we discuss the qualitative physics underlying this switch and its experimental relevance, and show that extension arises when the motor run-length and unbinding rate are relatively large compared to the filament length and the crosslink unbinding rate, respectively. We quantitatively compare the resulting tensions to those expected from simple mechanical nonlinearity and motor dwelling models in Sec. 'Quantitative aspects and alternative models' . We find that self-organization dominates in tightly connected actomyosin bundles, where filament buckling is hampered, and that it outcompetes dwelling in microtubule systems where the filaments are long enough for end-dwelling to be a rare occurrence. Finally, we discuss the conceptual implications of these simple, widely applicable ideas for the understanding of self-organization in active filament-motor systems.

## Model

We consider a bundle of polar filaments of length $L$ aligned in the $x$-direction and subjected to periodic boundary conditions. The filaments are rigid, ruling out contraction arising from mechanical nonlinearities (*Lenz et al., 2012a*). The motor velocity does not depend on its position on the filament, ruling out contraction arising from end-dwelling (*Liverpool and Marchetti, 2005*). A filament may point in the direction of positive or negative $x$, and maintains this polarity throughout the dynamics. At steady-state, filaments constantly grow from their plus ends and shrink from their minus ends at a fixed velocity $v_t$, a phenomenon known as 'treadmilling' in actin throughout which their length remains constant (*Alberts et al., 2015*; *Figure 2a*). While the dynamics of microtubules

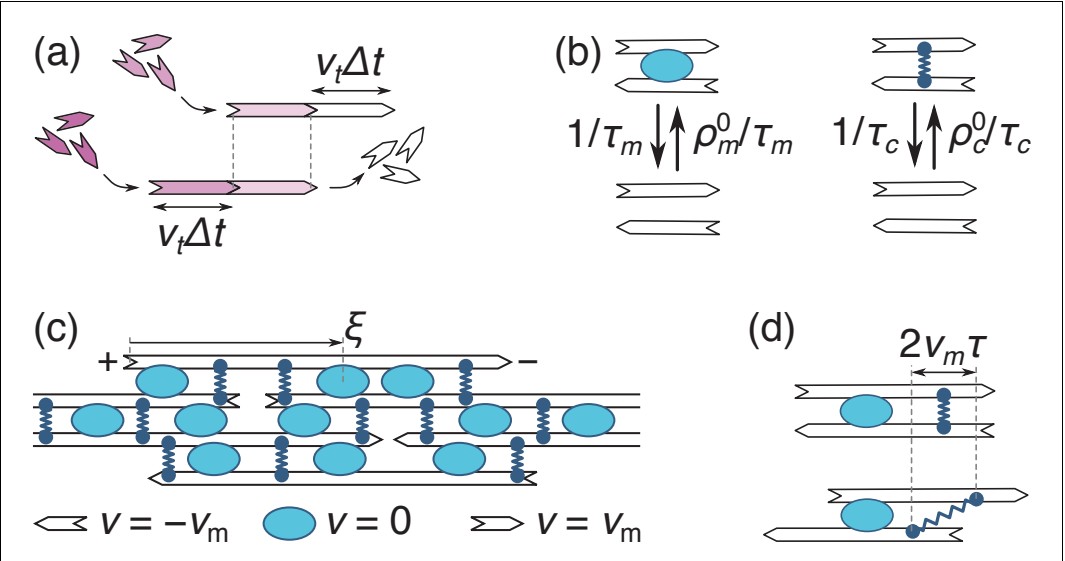

**Figure 2.** Principles of filament, motor and cross-link dynamics. (a) Simultaneous polymerization at the plus end (incoming purple monomers) and depolymerization from the minus end (outgoing white monomers) induce a leftwards 'treadmilling' motion of the filament. The top and bottom images respectively correspond to times $t$ and $t + \Delta t$. (b) Motors come on and off a pair of filaments with constant rates (on the left), and so do crosslinks (on the right). (c) In an assembly of identical filaments of mixed polarities where motors slide with a velocity $v_m$, a right-(left-)pointing filament moves with a velocity $v_m$ ($-v_m$) relative to any motor. Note that the coordinate $\xi$ is measured from the filament's *plus* end. (d) Crosslinks that remain bound to two antiparallel filaments throughout this dynamics stretch with a velocity $2v_m$ (the top and bottom panels represent the same system with a time interval $\tau$).

proceeds through somewhat different mechanisms, the insights gained from this simple, analytically tractable dynamics are also applicable there, as further described in the Discussion. Motors and crosslinks constantly bind and unbind from filaments, and we denote by $\tau_m$ ($\tau_c$) and $\rho_m^0$ ($\rho_c^0$) the average motor (crosslink) unbinding time and equilibrium density (*Figure 2b*). Finally we neglect viscous drag, as further argued in the Discussion.

Once bound to a filament, motors slide towards its plus end with a velocity $v_m$. The value of $v_m$ is set by a competition between the propulsive forces of the motors and the restoring forces of the crosslinks, and is to be determined self-consistently at a later stage of the calculation. In a mean-field description (valid for filaments interacting with many neighbors through many motors and crosslinks), this results in the pattern of motion illustrated in *Figure 2c*.

Focusing on a single right-pointing filament, the combined effect of motor motion and treadmilling implies that motors move with a velocity $v_m - v_t$ relative to the growing plus end. Denoting by $\xi$ the distance between the motor and the plus end (*Figure 2c*), this implies that the number of bound motors per unit filament length $\rho_m(\xi, t)$ satisfies the reaction-convection equation:

$$\partial_t \rho_m = -\partial_\xi J_m + \frac{\rho_m^0}{\tau_m} - \frac{\rho_m}{\tau_m}, \tag{1}$$

where $J_m(\xi, t) = \rho_m(v_t - v_m)$ is the motor current in the reference frame of the plus end, and $\rho_m^0/\tau_m$ represents the attachment rate of unbound motors from the surrounding solution. Newly polymerized filament sections in $\xi = 0$ do not yet have any motors bound to them, implying $\rho_m(0, t) = 0$ if $v_t > v_m$; likewise $\rho_m(L, t) = 0$ if $v_t < v_m$. Motors bound to two filaments of opposing polarities exert forces on each filament, and we denote by $f_m(\xi, t)$ the longitudinal force per unit length exerted by the motors on a right-pointing filament. For independent motors operating close to their stall force (i.e., motors whose velocity is essentially controlled by the external crosslink restoring forces), $f_m(\xi, t)$ is the ratio between the stall force of a single motor and the spacing between motors along the filament. It is thus proportional to the local motor density through $f_m(\xi, t) = f_m^0 \times [\rho_m(\xi, t)/\rho_m^0]$. In the

opposite limit where crosslinks are absent, motors slide filaments along at their unloaded velocity without producing any stresses, which rules out both contraction and extension (*Lenz et al., 2012a*). Note that motors do not induce internal forces in pairs of filaments with identical polarities, which we thus need not consider here.

The density $\rho_c(\xi, \tau, t)$ of crosslinks of age $\tau$ bound in $\xi$ at time $t$ satisfies the conservation equation

$$\partial_t \rho_c + \partial_\tau \rho_c = -\partial_\xi J_c + \frac{\rho_c^0 \delta(\tau)}{\tau_c} - \frac{\rho_c}{\tau_c}, \tag{2}$$

with $\rho_c(0, \tau, t) = \rho_c(\xi, \tau \leq 0, t) = 0$. Since the crosslink attachment points do not slide on the filament, their advection relative to the plus end is entirely due to treadmilling and the crosslink current reads $J_c(\xi, t) = \rho_c v_t$. The term $\partial_\tau \rho_c$ in *Equation 2* can be viewed as an advection term along the coordinate $\tau$, which accounts for the fact that the age $\tau$ of a bound crosslink increases linearly with time $t$. While attached crosslinks are thus advected towards increasing $\tau$, newly attached crosslinks all have age $\tau = 0$ by definition, which we enforce through the delta function in the source term $\rho_c^0 \delta(\tau) / \tau_c$. As motor forces tend to slide filaments of opposing polarities respective to one another, they are opposed by the restoring forces of the crosslinks, which tend to keep filaments stationary with respect to one another. To describe this competition, we assimilate crosslinks to Hookean springs with elastic constant $k_c$. The average extension of a crosslink bound to two antiparallel filaments is equal to zero at the time of its binding (denoted as $\tau = 0$), but increases as $2v_m\tau$ as the filaments slide respective to one another (*Figure 2d*). As each crosslink exerts a Hookean force $-k_c \times (2v_m\tau)$ on the filament, the crosslink force per unit filament length is obtained by summing this force over all filament ages, yielding $f_c(\xi, t) = \int_0^{+\infty} -k_c \times (2v_m\tau) \times \rho_c(\xi, \tau, t) \, \mathrm{d}\tau$.

## Results

### Self-organization and force distribution

Solving *Equations 1-2*, we compute the steady-state force densities exerted by the motors and crosslinks on the filament:

$$f_m(\xi) = \begin{cases} f_m^0 \left[ 1 - e^{-\xi/(v_t - v_m)\tau_m} \right] & \text{if } v_t > v_m \\ f_m^0 \left[ 1 - e^{-(L - \xi)/(v_m - v_t)\tau_m} \right] & \text{if } v_t < v_m \end{cases} \tag{3a}$$

$$f_c(\xi) = -2k_c \rho_c^0 \tau_c v_m \left[ 1 - \left( 1 + \frac{\xi}{v_t \tau_c} \right) e^{-\xi/v_t \tau_c} \right]. \tag{3b}$$

*Equation 3* describe a depletion of motors and crosslinks close to the filament ends, with associated depletion lengths $|v_t - v_m|\tau_m$ and $v_t\tau_c$, as illustrated in *Figure 3*. Similar nonuniform motor distributions have previously been studied to explain the length-dependence of microtubule depolymerization rates (*Varga et al., 2006*; *Reese et al., 2011*). The crosslink depletion results from the finite time required to decorate newly polymerized filament sections with crosslinks, while the motor depletion arises from the time required to dress a newly created filament overlap with motors. Provided the filament length is much larger than these depletion lengths, the motor force and crosslink friction asymptotically go to the constant values $f_m^0$ and $-2k_c \rho_c^0 \tau_c$ far from the filament ends as the motor and crosslink densities go to their equilibrium values.

### Velocity selection and bundle tension

To understand how the motor velocity $v_m$ is selected, we define $v_m^0 = f_m^0 / (2k_c \rho_c^0 \tau_c)$ as the speed at which the asymptotic forces $f_m^0$ and $-2k_c \rho_c^0 \tau_c v_m$ balance each other. The velocity $v_m^0$ thus characterizes the hypothetical motion of infinite-length filaments, where the effects of depletion are negligible. By contrast, shorter filaments undergo both a smaller overall driving force and a smaller friction. Depletion thus affects the velocity $v_m$, while $v_m$ itself affects motor depletion as described by *Equation 3a*. Here we analyze this mutual dependence for finite-length filaments.

Rescaling all lengths by $v_m^0 \tau_m$ and times by $\tau_m$, we henceforth denote dimensionless variants of previously introduced variables with a tilde and determine $\tilde{v}_m$ by demanding that the total force

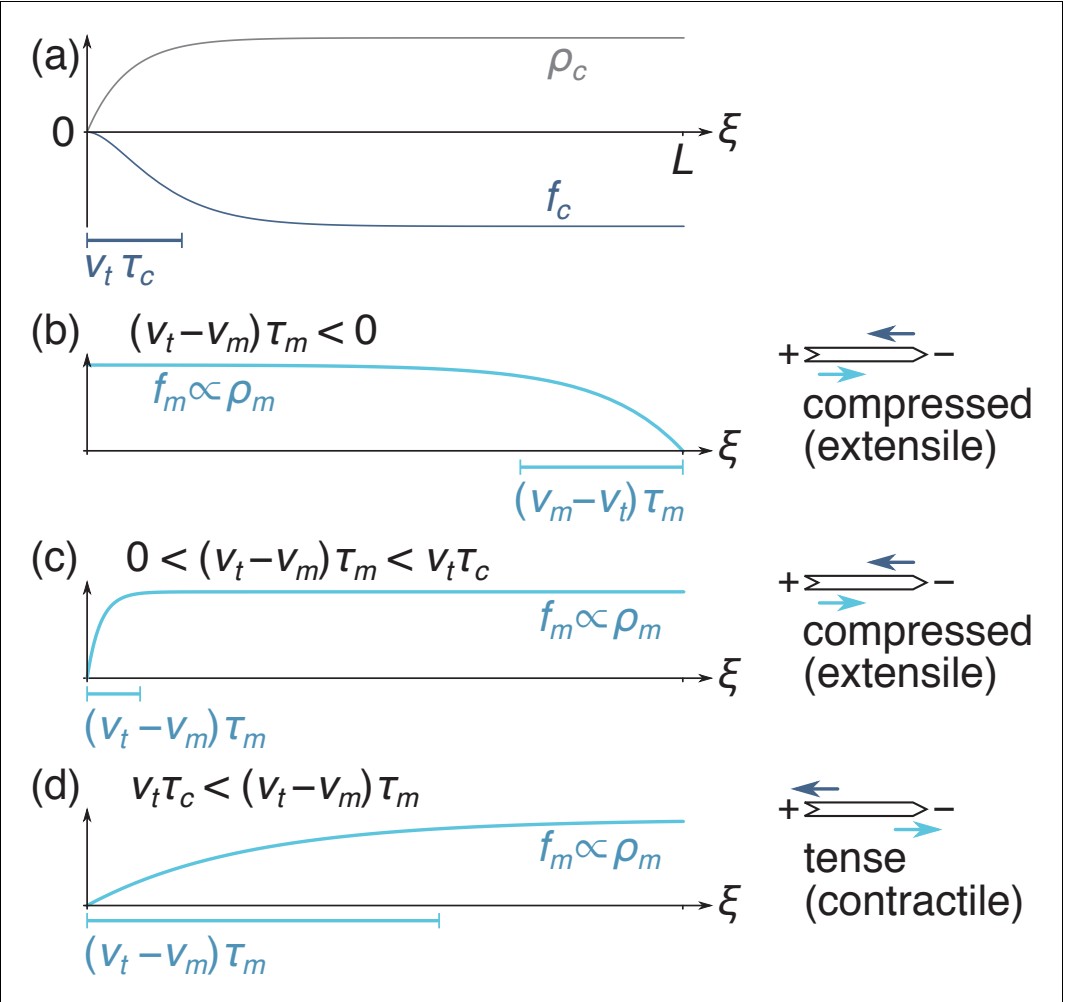

**Figure 3.** Filament force density profiles as in *Equation 3*. (a) The crosslink density $\rho_c$ is suppressed near the plus end, implying that the crosslink force $f_c<0$ is predominantly applied to the right-hand-side of the filament. (b) When motors are faster than treadmilling ($v_m>v_t$), they are depleted from the minus end and right-directed motor forces are predominantly applied on the left-hand side of the filament. As schematized on the right-hand-side, the fact that the crosslink force (dark blue arrow) is applied more to the right than the motor force (light blue arrow) implies that the filament is under compression. (c) When $v_t$ barely exceeds $v_m$, motor forces are applied relatively uniformly over the filament, which also results in filament compression. (d) When $v_t \gg v_m$, the motor depletion zone is larger than the crosslink depletion zone and motor forces are significantly shifted to the right. The filament is tensed in that case.

$F = \int_0^L [f_m(\xi) + f_c(\xi)]\,\mathrm{d}\xi$ exerted on a single filament vanishes. Defining $u = (\tilde{v}_t - \tilde{v}_m)/\tilde{L}$, we insert *Equation 3* into this condition and obtain a transcendental equation for $u$:

$$|u|(1 - e^{-1/|u|}) = (1-a) + bu, \tag{4}$$

where $a = \tilde{v}_t[1 - g(\tilde{v}_t\tilde{\tau}_c/\tilde{L})]$ and $b = \tilde{L}[1 - g(\tilde{v}_t\tilde{\tau}_c/\tilde{L})]$ are two constants and $g(y) = 2y - (1+2y)e^{-1/y}$ (see *Figure 4a*). As $a>0$ and $b>0$, *Equation 4* gives rise to three regimes illustrated in *Figure 4b–c*: one where translocation by the motors is faster than treadmilling ($u<0 \Leftrightarrow v_m>v_t$), one where treadmilling is faster than translocation ($u>0$) and one where one $u<0$ solution coexists with two $u>0$ solutions. We determine the stability of these solutions by perturbing $\tilde{v}_m$ by a small quantity $\delta\tilde{v}_m$ and assessing whether the overall force $F$ exerted on the filament tends to amplify or suppress this perturbation. We find that all unique solutions are stable (*i.e.*, $\partial F/\partial\tilde{v}_m<0$). In the three-solutions regime, the smaller of the two $u>0$ solutions is unstable. The bundle thus chooses one of the other two, resulting in two

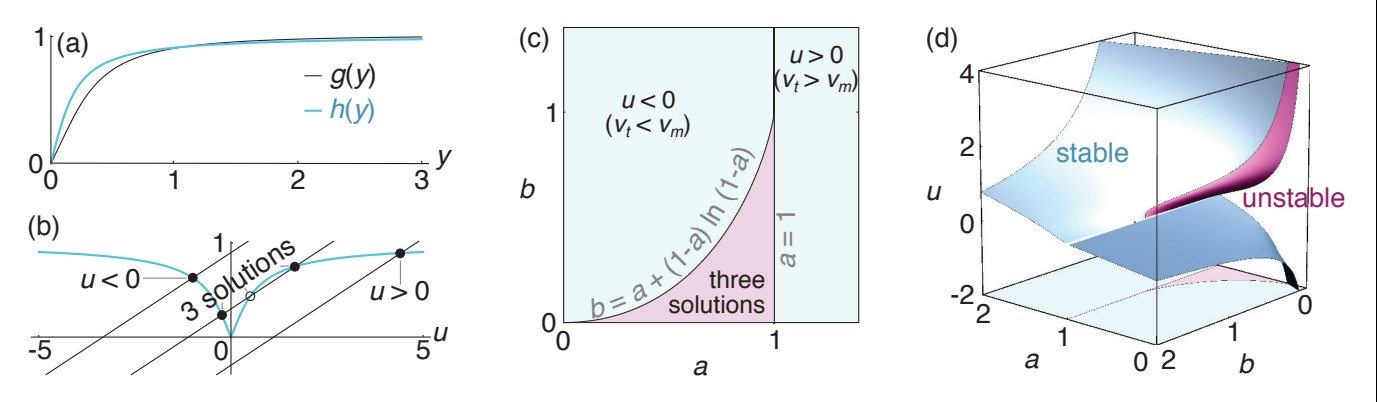

**Figure 4.** Velocity selection in the bundle. (a) Profiles of the functions $g(y)$ and $h(y)$, both of which go monotonically from 0 to 1 as $y$ goes from 0 to $+\infty$. (b) Graphical illustration of the velocity selection condition **Equation 4** as the intersection between two curves. The blue curve represents the left-hand side of **Equation 4**, and the black lines represent three possible parameter regimes for the right-hand side (here b = 0.27 and a = 0.1, 0.7 and 1.3 from left to right). Solid (open) circles represent stable (unstable) solutions. (c) Phase diagram presenting these three regimes as a function of parameters $a$ and $b$. (d) Values of the rescaled apparent filament velocity $u$ selected by the system, with colors indicating the stability of the solutions. The phase diagram of panel (c) is reproduced on the bottom face of the plot to facilitate comparisons.

coexisting stable solutions of opposing signs as illustrated in **Figure 4d**. As for any first-order (discontinuous) transition, bundles in this parameter regime will select either value of $u$ depending on their initial condition, and any switching from one to the other involves hysteresis.

We now turn to the contractile/extensile character of a bundle comprised of $\rho_f$ filaments per unit length. A filament in this bundle is subjected to a total force per unit length $f(\xi) = z[f_m(\xi) + f_c(\xi)]$ at location $\xi$, where $z$ denotes the number of interacting antiparallel neighbors of a filament. As the filament tension $T(\xi)$ vanishes at the filament ends $[T(0) = T(L) = 0]$, its tension in $\xi$ thus reads $T(\xi) = -\int_0^\xi f(\xi')\,\mathrm{d}\xi'$. The contractile or extensile character of our bundle is revealed by its integrated tension across any $x = \mathrm{constant}$ plane. In thick bundles, this plane is intersected by a large number of filaments (namely $\rho_f L \gg 1$) each intersecting the plane at a random coordinate $\xi$ that is uniformly distributed between 0 and $L$. As a result, the bundle tension is given by the average $\mathcal{T} = \rho_f \int_0^L T(\xi)d\xi$. Defining $\tilde{\mathcal{T}} = \mathcal{T}/(z\rho_f L^2 f_m^0) = \tilde{\mathcal{T}}_m + \tilde{\mathcal{T}}_c$, the respective contributions of the motors and crosslinks to the dimensionless bundle tension are

$$\tilde{\mathcal{T}}_m = \begin{cases} \frac{1}{2} - u^2 + u(1+u)e^{-1/u} & \text{if } u > 0 \\ \frac{1}{2} - u^2 e^{1/u} + u(1+u) & \text{if } u < 0 \end{cases} \tag{5a}$$

$$\tilde{\mathcal{T}}_c = \frac{|u|\left(1 - e^{-1/|u|}\right) - 1}{4}\left[2 + h\left(\frac{\tilde{v}_t \tilde{\tau}_c}{\tilde{L}}\right)\right], \tag{5b}$$

where the function $h(y) = [4y - 12y^2 + (2 + 8y + 12y^2)e^{-1/y}]/[(1 - 2y) + (1 + 2y)e^{-1/y}]$ is illustrated in **Figure 4a**. As shown in **Figure 5a**, these expressions can result in either sign for $\mathcal{T}$ depending on the values of $u$ and $h(\tilde{v}_t \tilde{\tau}_c/\tilde{L})$. As the periodic boundary conditions used here confine the bundle to a fixed length, a bundle with a propensity to extend develops a negative tension $\mathcal{T} < 0$ (i.e., is compressed), while $\mathcal{T} > 0$ denotes a contractile (tense) bundle. These two behaviors respectively correspond to the situations illustrated in **Figure 3b–c** and **Figure 3d**.

## Qualitative predictions

To analyze the different regimes accessible to our bundle, we illustrate them in **Figure 5b** as a function of the original dimensionless parameters $\tilde{v}_t$, $\tilde{\tau}_c$ and $\tilde{L}$. As some parameter values yield coexisting metastable solutions for $u$, so can they allow for both contractile and extensile steady states. However, despite this ambiguity at intermediate parameter values, **Figure 5b** shows that the self-

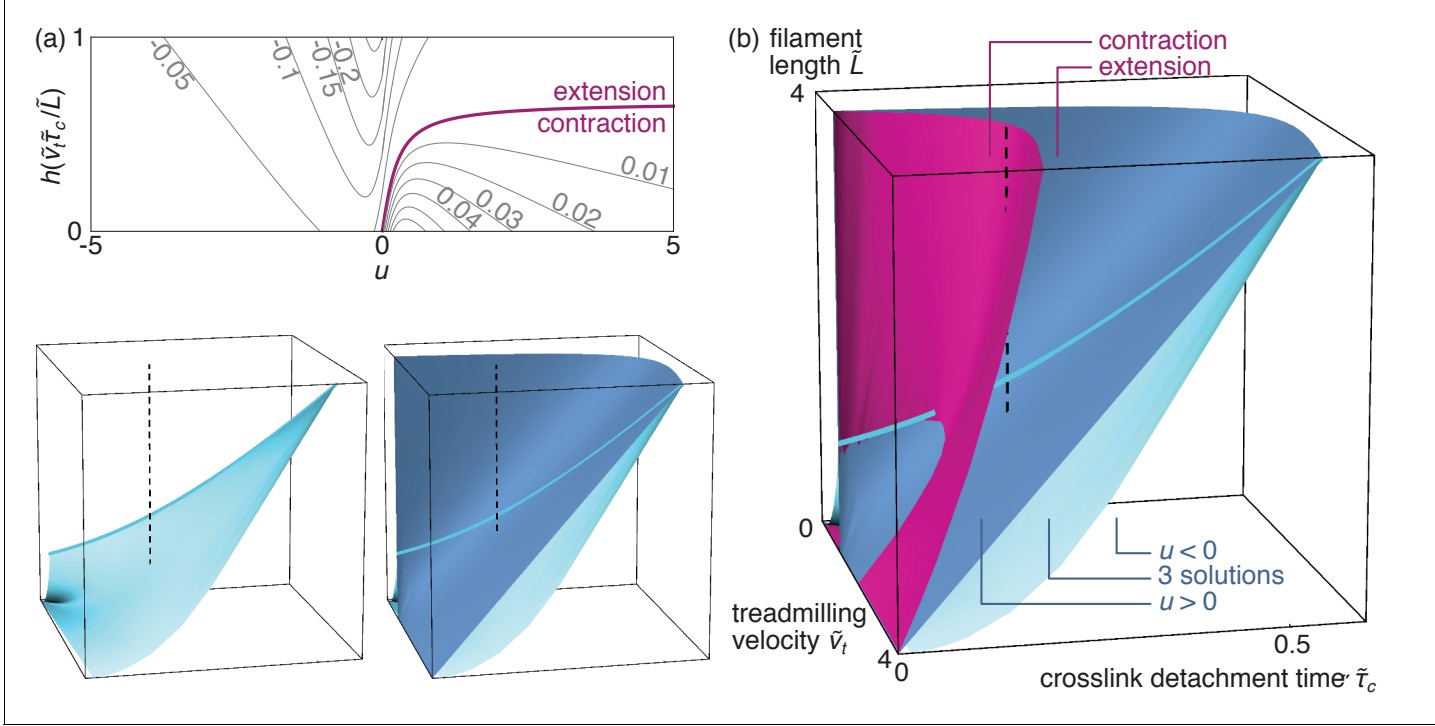

**Figure 5.** Bundle tension. (a) Level curves for the dimensionless bundle tension $\tilde{\mathcal{T}}$ as a function of the apparent velocity $u$ determined from *Equation 4* and pictured in *Figure 4d*, and the ratio $\tilde{v}_t \tilde{\tau}_c / \tilde{L}$. The $\tilde{\mathcal{T}} = 0$ purple line separates contraction from extension. (b) Contraction regimes associated with the stable $u > 0$ solution (purple surface) and velocity regimes as in *Figure 4c* (blue surfaces) as a function of the dimensionless parameters $\tilde{v}_t$, $\tilde{\tau}_c$ and $\tilde{L}$. The blue surfaces are plotted separately on the bottom left to facilitate visualization (axes are as in the main figure), and cuts through the 3D diagram are shown in the appendix. As discussed above the 'three solutions' regime comprised between these two surfaces has coexisting stable $u < 0$ and $u > 0$ solutions. The light blue line outlines the intersection between the two blue surfaces. The dashed line materializes one set of reasonable experimental parameters (see text), and goes from $u < 0$ to $u > 0$ through the coexistence ('three solutions') region, implying a first-order transition. By contrast, a similar vertical line shifted to smaller values of $\tilde{v}_t$ would describe a second-order transition.

organization mechanism investigated here results in unambiguous extension for broad ranges of parameters.

To understand this reversal of contractility qualitatively, we first consider a contractile situation at low crosslink detachment time $\tau_c$. From there, increasing $\tau_c$ results in an enlarged crosslink depletion zone in the vicinity of the filament plus ends (*Figure 3*), and thus in a relative localization of the crosslinks towards the filament minus ends. This 'anti-sarcomere' organization results in an extensile bundle (*Figure 1c*, right) at large $\tau_c$, in contrast with the contractile 'sarcomere' structures (*Figure 1c*, left) found in our highly organized striated muscle.

The extension mechanism reported here differs from behaviors previously modeled in the theoretical literature, both in its cause and in its applicability. Extension has indeed been observed in numerical simulations reported in *Belmonte et al. (2017)*, but only in situations where the motors were assumed to dwell at the filament ends. The discussion section of this reference does however qualitatively predict that extension should result from an 'antenna' mechanism similar to the depletion quantitatively modeled here. Another instance of bundle extension is reported in *Kruse and Sekimoto (2002)*. In that model, the extensile behavior stems from the interaction between *parallel*, not antiparallel filaments, which is at odds with experimental observations that bundle-wide force generation in actomyosin requires antiparallel filaments (*Reymann et al., 2012*). Finally, in two-dimensional numerical simulations extensile antiparallel microtubule pairs were reported due to the fact that they typically bind halfway between the two configurations of *Figure 1b*, implying that their subsequent dynamics predominantly involves extensile configurations (*Gao et al., 2015*). The freedom required for this mid-filament binding is however likely not afforded to tightly bundled

filaments, although the non-uniform motor distributions discussed here could also have contributed to the extensile behavior observed in *Gao et al. (2015)*.

Beyond the transition from contraction to extension, *Figure 5b* shows that a second transition can be triggered by a further increase of $\tau_c$ in the extensile phase, which causes the variable $u \propto v_t - v_m$ to change sign through a first-order (for small $\tilde{L}$) or a second-order (for large $\tilde{L}$) transition (*Figure 5b*). Indeed, the enhanced crosslink depletion associated with a large $\tau_c$ tends to reduce the friction between filaments, resulting in faster motor motion and thus in a situation where motor sliding outpaces treadmilling ($u<0$).

These two transitions could be observed in vitro, and possibly even in vivo. Indeed, the contractile *vs.* extensile character of actomyosin bundles is apparent from the direct imaging of reconstituted assays (*Thoresen et al., 2011*; *Reymann et al., 2012*) as well as cells (*Mendes Pinto et al., 2012*). The second, velocity-reversal transition, on the other hand, can be monitored in experiments where single filaments are resolved (*Murrell and Gardel, 2012*). As motor velocity outpaces treadmilling, such filaments will switch from an apparently plus-end-directed motion (illustrated in *Figure 2a*) to a motor-induced, minus-end-directed motion. In practice, these transition could be induced in a number of ways, including changes in the monomeric actin concentration or the action of formin (affecting $L$ and $v_t$), the presence of different types of crosslinks (affecting $v_m^0$ and $\tau_c$), or modifications of the number or type of motor heads in a thick filament (affecting $v_m^0$ and $\tau_m$) (*Thoresen et al., 2013*). Such changes could also be at work in smooth muscle, where the number of myosins in individual thick filaments is regulated dynamically (*Seow, 2005*). The experimental relevance of these transitions is illustrated by a dashed line in *Figure 5b*, which shows that both transitions can be probed by varying $L$ between 250 nm and 1 μm while holding $v_m^0 = 50\,\mathrm{nm} \cdot \mathrm{s}^{-1}$, $\tau_m = 5\,\mathrm{s}$ (*Erdmann et al., 2013*), $v_t = 100\,\mathrm{nm} \cdot \mathrm{s}^{-1}$ (*Howard, 2001*), and $\tau_c = 1\,\mathrm{s}$ (*Miyata, 1996*) fixed.

## Quantitative aspects and alternative models

The magnitude of the forces and velocities predicted by our model are on par with those found in vivo, for example in the cytokinetic ring of fission yeast. Indeed, setting $L = 1.4\,\mu\mathrm{m}$, $\rho_f L = 20$, $f_m^0 \simeq 7.2 \times 10^{-6}\,\mathrm{N} \cdot \mathrm{m}^{-1}$ (*Wu and Pollard, 2005*), $k_c \simeq 3 \times 10^{-4}\,\mathrm{N} \cdot \mathrm{m}^{-1}$ (*Rief et al., 1999*), $z = 3$ as in a hexagonal packing of alternating left- and right-pointing filaments and $\tilde{\mathcal{T}} \simeq 0.2$, we find a contractile force $\mathcal{T} \simeq 120\,\mathrm{pN}$ comparable with the ring tension of 390 pN measured in fission yeast protoplasts (*Stachowiak et al., 2014*) We also find a characteristic velocity $v_m^0 \simeq 5\,\mathrm{nm} \cdot \mathrm{s}^{-1}$ similar to that of ring contraction ($\simeq 3-4\,\mathrm{nm} \cdot \mathrm{s}^{-1}$). These order of magnitudes retrospectively justify our choice to neglect viscous drag forces in our system, which are of order of $\eta L v_m^0 \simeq 7 \times 10^{-18}\,\mathrm{N}$, where $\eta \simeq 10^{-3}\,\mathrm{Pa} \cdot \mathrm{s}^{-1}$ is the viscosity of water. Note also that other mechanisms for contractility based on sarcomere-like crosslinking of the filament barbed ends have also been proposed in the specific case of fission yeast (*Thiyagarajan et al., 2017*).

In this section, we show that in addition to being on par with experimentally observed forces, the tensions generated by our self-organization mechanism exceed those resulting from mechanical non-linearities (*e.g.*, buckling) or motor dwelling over broad ranges of parameters, suggesting that the self-organization mechanism could be a substantial contributor to bundle tension in vitro and in vivo.

### Self-organization *vs.* buckling-induced tensions

Buckling-induced contractility is relevant for bundles comprising flexible (typically actin) filaments. To estimate the associated tension, we use the model of *Lenz et al. (2012b)*, where filaments may locally buckle over a small section bounded by a crosslink and a motor (*Figure 6a*). Filaments retain their overall linear shape outside of the buckled regions, and it is thus reasonable to assume that self-organization and buckling can operate simultaneously and that they contribute additively to the bundle tension. Here we compare the tension $\mathcal{T}_{\mathrm{buckling}}$ induced by the latter mechanism to the tension $\mathcal{T}$ given by *Equation 5* as a function of two experimentally adjustable parameters, namely the number $n$ of myosin heads per myosin minifilament (*Thoresen et al., 2013*), and the average spacing $\ell_0$ between two consecutive motors or crosslinks (*Lenz et al., 2012b*).

To compute the tensions of interest, we extrapolate their value to a bundle with one filament per cross-section, keeping in mind that both $\mathcal{T}$ and $\mathcal{T}_{\mathrm{buckling}}$ scale linearly with that number in thicker bundles. To obtain the value of $\mathcal{T}_{\mathrm{buckling}}$, we first note that buckling-induced force generation in an

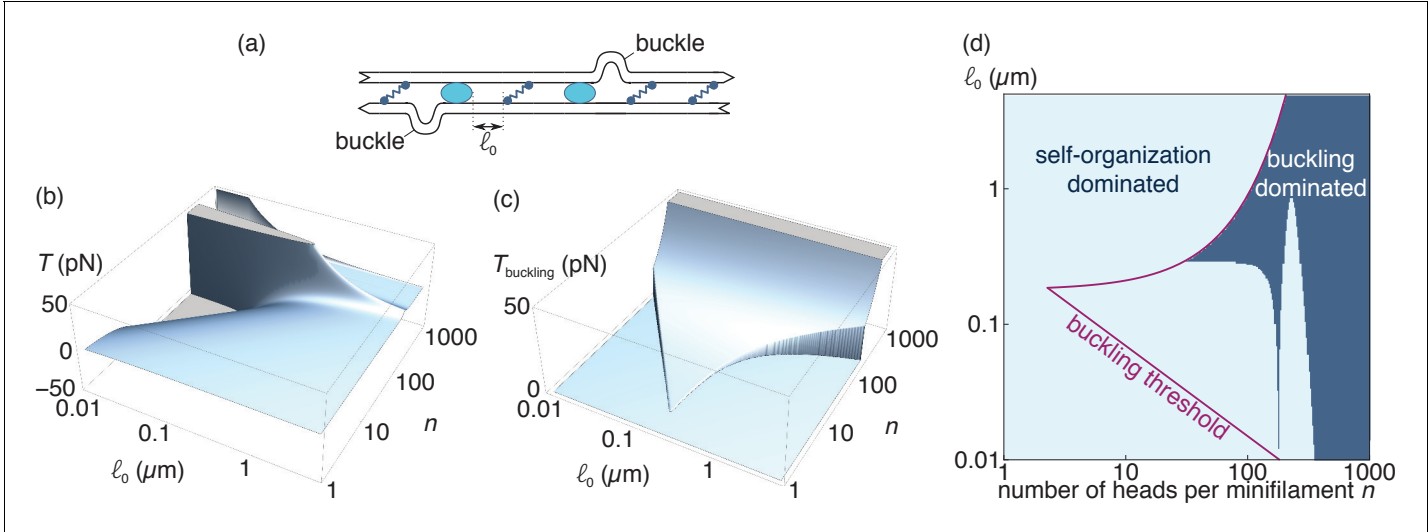

**Figure 6.** Comparison of the self-organization and buckling mechanisms (relevant for actin). (a) At length scales of the order of the distance between consecutive motors and crosslinks (*i.e.*, much smaller than those considered in the main text), filament buckling can generate an additional source of bundle contraction (*Lenz et al., 2012a*) (b). Specifically, the random juxtaposition of motors and crosslinks within a bundle create regions of alternating compressive and extensile stresses along individual filaments, which can result in filament buckling if the compressive forces exceed the buckling threshold of a filament section. Following buckling, the bent part of the filament becomes essentially irrelevant elastically and the tensile force $F_s$ exerted by the motor on the non-buckled filament sets the tension of the bundle. (b) Tension per filament induced by the self-organization mechanism and (c) the buckling mechanism. Note that $\mathcal{T}$ can take both positive and negative values, denoting contraction and extension respectively, while $\mathcal{T}_{\text{buckling}}$ is always positive. We set $\mathcal{T}_{\text{buckling}}$ to zero outside of the interval $\ell_0^- < \ell_0 < \ell_0^+$ where buckling is allowed. (d) Bounds of the interval where buckling is allowed (*purple line*) and the parameter regimes where $|\mathcal{T}| > \mathcal{T}_{\text{buckling}}$ (*light blue region*).

actomyosin bundle requires that the typical compressive forces on a filament exceed the buckling threshold of that filament. As shown in *Lenz et al. (2012b)*, this requires that the typical spacing $\ell_0$ between two motors/crosslinks lies between the following two bounds:

$$\ell_0^- = (k_B T \ell_p / F_s L^{1/2})^{2/3} \tag{6a}$$

$$\ell_0^+ = (L v \tau_m \ell_p)^{2/5}, \tag{6b}$$

where $k_B T$ is the thermal energy, $\ell_p$ the persistence length of a filament, $L$ its actual length, $F_s$ is the stall force of a single motor, $v$ its unloaded velocity and $\tau_m$ its detachment time. Qualitatively, the condition $\ell_0 > \ell_0^-$ accounts for the requirement that the filaments be long enough (and thus floppy enough) for the motor forces to be able to buckle them, while the upper bound $\ell_0 < \ell_0^+$ expresses the fact that building up a force sufficient to buckle a filament in bundles where motors and crosslinks are far apart takes so long that spontaneous motor detachment will hinder it. Following *Rosenfeld et al. (2003)* and *Lenz et al., 2012b*, we use $\tau_m = 0.96^n \times 3 \, \text{ms}$, as well as $F_s = n \times 0.1 \, \text{pN}$, where the low value of the stall force per myosin head accounts for their intermittent attachment to the filaments. We also set $\ell_p = 10 \, \mu\text{m}$, $L = 5 \, \mu\text{m}$ and $v = 200 \, \text{nm} \cdot \text{s}^{-1}$ as in *Lenz et al. (2012b)*. In situations where the condition $\ell_0^- < \ell_0 < \ell_0^+$ is satisfied, we set the contractile force of a buckled bundled to $F_s$, consistent with the idea that buckling actin essentially mechanically removes the compressed sections of the filament, allowing individual motors to act as if in a sarcomeric configuration.

Complementing these assumptions with the values of $v_t$, $v_m^0$, $\tau_c$ and $z$ used in Sec. 'Qualitative predictions', we plot the tension per filament induced by either mechanism in *Figure 6b–c*, and directly compare them in *Figure 6d*. Self-organization trivially dominates outside of the region where buckling is allowed, which represents a substantial fraction of the reasonably accessible parameter regimes. Moreover, even in the region where buckling is allowed self-organization dominates for small values of $\ell_0$ as long as $n$ does not become very large. These results support the notion that

positional self-organization of motors and crosslinks constitutes a viable mechanism for actomyosin force generation despite the possibility of buckling-induced contraction.

## Self-organization *vs.* dwelling-induced tensions

Motor dwelling at the filament ends takes place in certain types of microtubule-associated motors (*Roostalu et al., 2018*; *Tan et al., 2018*), and could occur in some actin-myosin systems (*Wollrab et al., 2018*). To estimate the associated tension, we consider a variant of our self-organization model where motors reaching the plus end of a filament dwell there for an average time $\tau_m$. The motor attachment-detachment dynamics is infinitely fast everywhere else, and so is that of the crosslinks. This rules out the depletion effects discussed in Sec. 'Self-organization and force distribution'.

These assumptions imply uniform bulk densities of non-dwelling motors $\rho_m(\xi \neq 0, t) = \rho_m^0$ and crosslinks $\rho_c(\xi, t) = \rho_c^0$. In the case $v_m > v_t$, the flow of motors into the filament plus end is equal to $|J_m(\xi = 0)| = \rho_m(v_m - v_t)$, implying that at steady state an average number $\rho_m(v_m - v_t)\tau_m$ of motors dwells there. Assuming as before that motors operate at their stall force, this implies

$$f_m(\xi) = f_m^0 + f_m^0(v_m - v_t)\tau_m[\delta(\xi) + 1/L] \tag{7a}$$

$$f_c(\xi) = -2k_c\rho_c^0\tau_c v_m, \tag{7b}$$

where the first term of the right-hand side of *Equation 7a* accounts for the forces exerted on the filament by non-dwelling motors, while the second term incorporates the effects both of motors dwelling on the filament of interest (through the delta function) and on other filaments. Our description does not include doubly-dwelling motors, as the pattern of filament motion illustrated in *Figure 2c* implies that such filaments are immediately ripped from either one of the filaments they are attached to. In the case $v_m < v_t$, the motion of the motors is not fast enough to allow them to reach the plus end. Instead, the depolymerizing minus end catches up to them. This configuration has not been observed or proposed to lead to dwelling to my knowledge, and in the absence of dwelling no tension is generated.

To characterize the resulting bundle steady-states, we apply the velocity selection and stability criteria described in Sec. 'Velocity selection and bundle tension' and compute the resulting bundle tension. The case $\tilde{v}_t \geq 1$ implies that fast-depolymerizing minus ends catch up on the motors as discussed above, ruling out out dwelling and implying $\tilde{v}_m = 1$ and $\tilde{\mathcal{T}}_{\text{dwell}} = 0$. On the other hand, if $\tilde{v}_t < 1$ motors are faster than treadmilling and localize at the plus ends, inducing extension. In that case, the bundle finds a steady state provided that $\tilde{L} > 2$, with $\tilde{v}_m = (\tilde{L} - 2\tilde{v}_t)/(\tilde{L} - 2)$ and $\tilde{\mathcal{T}}_{\text{dwell}} = -(1 - \tilde{v}_t)/2(\tilde{L} - 2)$. Finally, if $\tilde{v}_t < 1$ and $\tilde{L} < 2$, no steady state exists in the system. To understand this, consider a situation where motors start accumulating at the filament plus end, increasing the propulsive force on the filament and thus increasing $v_m$, leading to a further increase in the number of accumulated motors. In the model, this positive feedback results in an infinite increase in velocity $v_m$ unless crosslink-induced friction stops it. If $\tilde{L} < 2$ however, filaments are very short, implying a small number of crosslinks and a comparatively small effective friction, hence the absence of a steady-state. In practice, such a situation is stabilized by effects ignored here, including the onset of motor depletion on the filament or a departure of the motors from their stall conditions.

As the dwelling and self-organization mechanisms both rely on a localization of the motors induced by the filament/motor dynamics, they result in quantitatively similar tensions as long as crosslink localization remains limited, that is for small values of $\tilde{\tau}_c$ (*Figure 7*). However, self-organization dominates over dwelling in situations where extended depletion profiles are allowed to develop, that is when crosslinks are long-lived or filaments are long (large $\tilde{\tau}_c$ or large $\tilde{L}$), as well as when fast treadmilling prevents motor dwelling ($\tilde{v}_t > 1$). We thus expect that the self-organization mechanism will be a substantial contributor to force generation in microtubule-motor systems not only in the obvious cases where motors do not dwell on the filament, but also in many situations where they do.

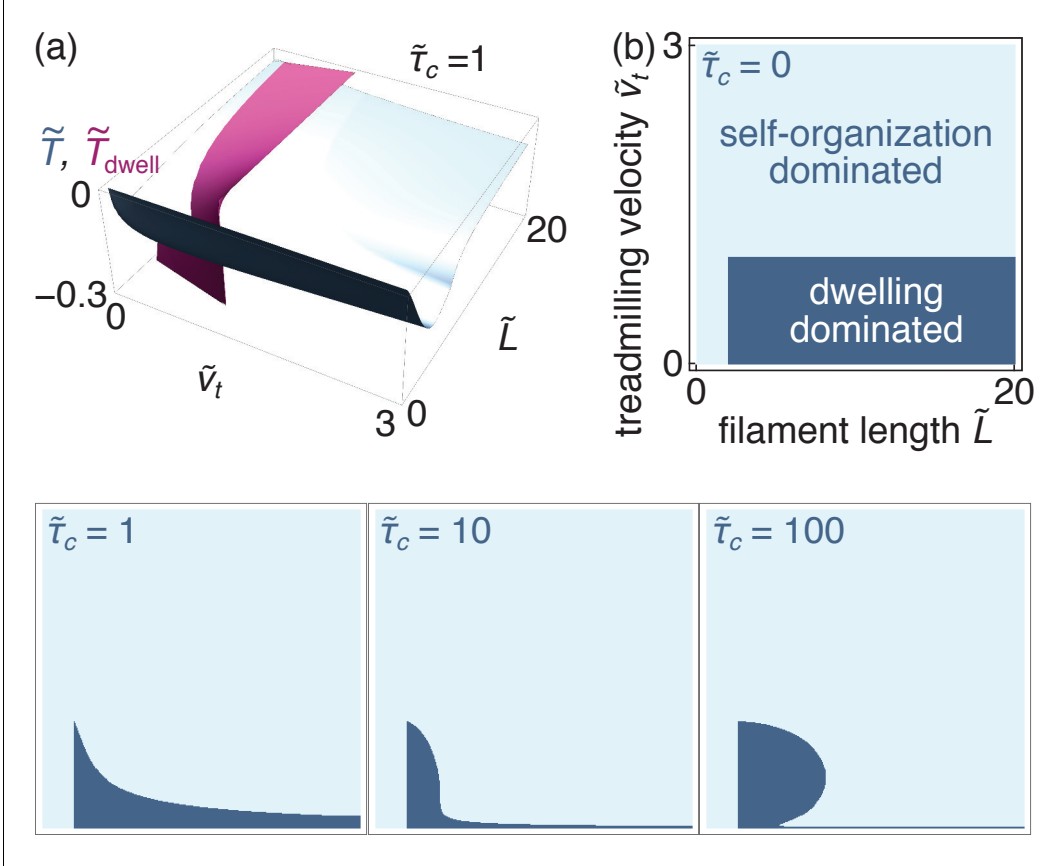

**Figure 7.** Comparison of the self-organization and dwelling mechanisms (relevant for microtubules). (a) Dimensionless tensions induced by the self-organization (*blue*) and the dwelling (*purple*) mechanisms for $\tilde{\tau}_c = 1$. (b) Parameter regimes where either regime dominates for different values of $\tilde{\tau}_c$ (all four panels use identical axes).

## Discussion

In contrast with the static organization of striated muscle, many non-muscle actomyosin structures as well as motor-microtubules assemblies are very dynamic, and their components continuously assemble and disassemble even as they exert forces on their surroundings. While numerical simulations are useful to investigate these systems (*Oelz et al., 2015*; *Kim, 2015*; *Ennomani et al., 2016*), the many parameters involved have until now hampered systematic explorations of all possible dynamical regimes. By contrast, our analytical approach allows us to derive a complete phase diagram for self-organized motor-filament bundles. *Lenz et al. (2012b)*, *Lenz (2014)* and *Belmonte et al. (2017)* derive similarly useful analytical results applicable to situations dominated by filament nonlinearity or motor dwelling. We thus uncover two previously unreported, experimentally observable transitions between bundle contraction and extension, and between plus-end-directed and minus-end-direction apparent filament motion. These transitions could serve as experimental signatures of self-organization-driven cytoskeletal force generation.

While some of our simplifying assumptions may affect the accuracy of our quantitative predictions, the simplicity of the underlying mechanisms make our qualitative statements very robust. Indeed, we predict extension whenever the filament plus end polymerizes quickly enough to induce significant crosslink depletion in its vicinity, resulting in a sarcomere-like organization. This depletion is insensitive to whether filament disassembly occurs through depolymerization at the minus end or cofilin-induced severing (*Theriot, 1997*). It is additionally relevant for dynamical microtubules, whose minus ends are mostly static while the plus end grows slowly before quickly retracting in a so-called 'catastrophe'. Our model will thus accurately predict the tension resulting from plus-end depletion during the growth phase, while fast catastrophes can be seen as more or less instantaneous filament deletion events without significant effect on bundle tension. Depletion is also present whether

crosslinks detach at a constant rate as assumed here, or unbind increasingly quickly under increasing force (*Miyata, 1996*). Turning to motors, we note that although small motor numbers may add significant density and velocity fluctuations to our mean-field model, motors are depleted on average in newer filament overlaps even in the presence of these fluctuations. While we describe this effect as a consequence of delayed motor binding from the surrounding solution, a similarly reduced force could also arise in bundles densely covered by motors, for instance due to a delay in fully aligning the myosin minifilament with the two antiparallel actin filaments to allow all myosin heads to fully participate in filament sliding. Finally, while we assume that motors always exert their stall force and thus acquire a velocity inversely proportional to the effective crosslink friction, introducing a more complicated motor force-velocity relationship would slightly complicate this dependence quantitatively, but not qualitatively.

Our prediction of a robust extensile regime provides a stringent test to validate or invalidate the self-organized force generation model in specific experiments. As such, it constitutes an important statement even for systems in which extension is *not* observed, as it implies that the absence of extension in certain parameter regimes argues against self-organization mechanisms in favor of mechanical nonlinearity (in actin) or motor dwelling (in microtubules) models. As an illustration, *Thoresen et al. (2011)* and *Thoresen et al. (2013)* report a setup where actomyosin bundles contract in the absence of treadmilling, in contrast with the prediction of *Figure 5b* that $v_t = 0$ implies extension. This discrepancy tends to disqualify self-organized contraction in this setup, and retrospectively validates the proposal made in *Lenz et al. (2012b)* that mechanical nonlinearities and specifically filament buckling dominate this assay. Conversely, self-organized force generation is likely to play a role in a number of in vivo actomyosin contractile structures where buckling is not observed (*Cramer et al., 1997*; *Kamasaki et al., 2007*) and where actin treadmilling dynamics plays a crucial role (*Mendes Pinto et al., 2012*). In microtubules, extension is observed consistent with our prediction when the filament polymerization/depolymerization dynamics is blocked in vitro (*Sanchez et al., 2012*; *Keber et al., 2014*), while both extension and contraction can arise in more complex in vivo situations (*Patel et al., 2005*; *Foster et al., 2015*). Both extension and contraction have also been reported in actin bundles, which cannot buckle due to their large stiffness (*Stam et al., 2017*).

Beyond steady-state contraction or extension, transitions between these two states could help understand several in vivo behaviors involving alternating contractions and expansions of the actomyosin cortex, including cell area oscillations observed during *Drosophila*, *C. elegans*, and *Xenopus* development (*Martin et al., 2009*; *Solon et al., 2009*; *Roh-Johnson et al., 2012*; *He et al., 2010*; *Kim and Davidson, 2011*; *Levayer and Lecuit, 2012*) or propagating actomyosin contractility waves (*Allard and Mogilner, 2013*). We speculate that such oscillations could arise through a Hopf bifurcation involving the rapid switching between a contractile and an extensile metastable state in the multiple-solution regime of *Figure 4*.

It would be interesting to see how the mechanisms described here apply to two- or three-dimensional actomyosin assemblies, whose richer geometry allows for additional actomyosin force generation mechanisms (*Lenz, 2014*). More refined approaches could also include discussions of the onset of positional ordering of the filaments themselves within the bundle (*Kruse et al., 2001*; *Kruse et al., 2003*; *Friedrich et al., 2012*). While such ordering is suppressed by filament diffusion (*Zemel and Mogilner, 2009*) and is not observed in many disordered actomyosin bundles (*Cramer et al., 1997*; *Kamasaki et al., 2007*), its onset during the formation of stress fibers is quite dependent on actin filament dynamics, suggesting a role for the mechanisms considered here (*Hu et al., 2017*). Finally, the fundamental principles for the dynamical depletion of motors and crosslinks described here could serve as guiding principles in our developing understanding of self-organized contractility in the cytoskeleton (*Nakazawa and Sekimoto, 1996*; *Kruse and Jülicher, 2000*; *Kruse and Sekimoto, 2002*; *Oelz et al., 2015*; *Belmonte et al., 2017*).

## Acknowledgements

I thank Alex Mogilner for sharing *Oelz et al. (2015)* before publication, Pierre Ronceray for enlightening discussions and Samuel Cazayus-Claverie, Michael Murrell, Guglielmo Saggiorato and Danny Seara for comments on the manuscript. This work was supported by Marie Curie Integration Grant PCIG12-GA-2012–334053, 'Investissements d'Avenir' LabEx PALM (ANR-10-LABX-0039-PALM), ANR

grant ANR-15-CE13-0004-03 and ERC Starting Grant 677532. My group belongs to the CNRS consortium CellTiss.

## Additional information

### Funding

| Funder | Grant reference number | Author |
| --- | --- | --- |
| FP7 People: Marie-Curie Actions | PCIG12-GA-2012-334053 | Martin Lenz |
| H2020 European Research Council | Stg677532 | Martin Lenz |
| LabEx PALM | ANR-10-LABX-0039- PALM | Martin Lenz |
| Agence Nationale de la Recherche | ANR-15-CE13-0004-03 | Martin Lenz |

The funders had no role in study design, data collection and interpretation, or the decision to submit the work for publication.

### Author contributions

Martin Lenz, Conceptualization, Formal analysis, Funding acquisition, Validation, Investigation, Visualization, Methodology, Project administration

### Author ORCIDs

Martin Lenz (iD) https://orcid.org/0000-0002-2307-1106

### Decision letter and Author response

Decision letter https://doi.org/10.7554/eLife.51751.sa1
Author response https://doi.org/10.7554/eLife.51751.sa2

## Additional files

### Supplementary files

• Transparent reporting form

### Data availability

This study does not involve the generation or analysis of data.

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

## Appendix 1

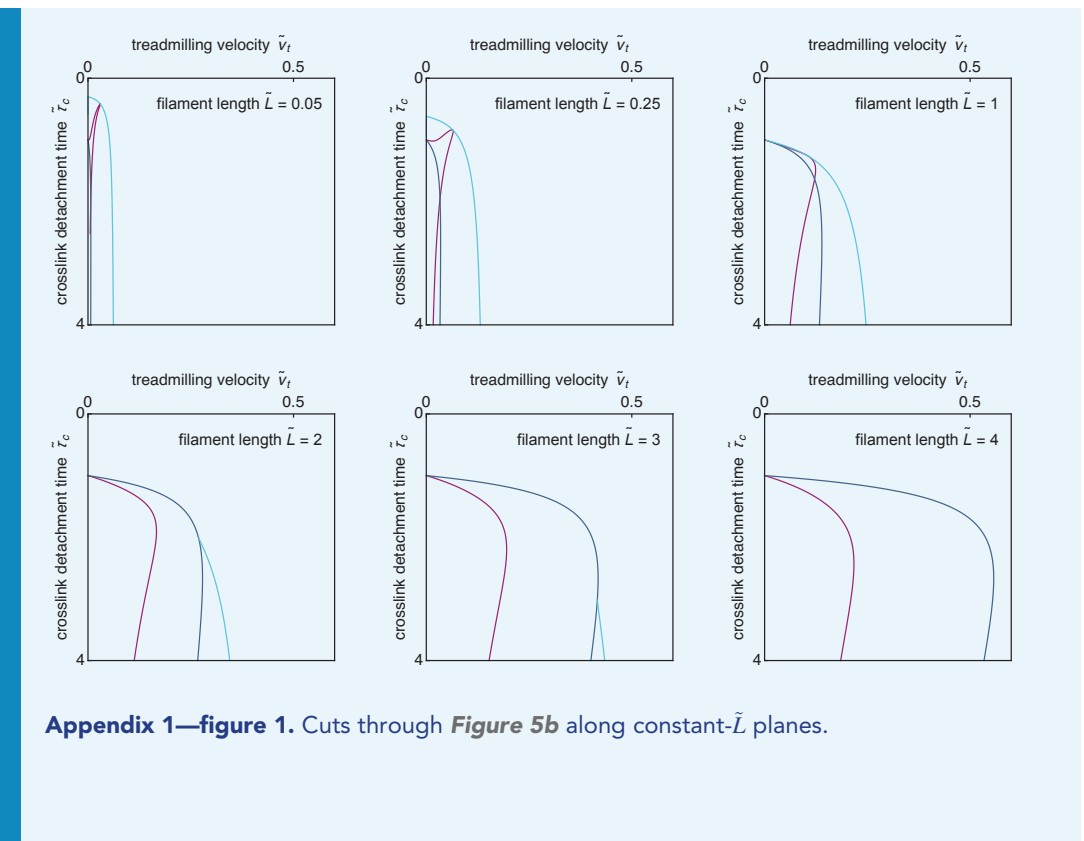

**Appendix 1—figure 1.** Cuts through *Figure 5b* along constant-$\tilde{L}$ planes.

