## [Decision Letter]

**Acceptance summary:**

This is a very nice theoretical paper on the collective dynamics of cytoskeletal bundles. The author critically examines several competing models for the competition between contraction and expansion of such bundles, and develops a systematic and very clear analysis that focuses on just one of the important issues, putting aside end-dwelling and filament buckling. He finds the possibility of reversal of contractility and argues that this supports the hypothesis of collective, self-organized behaviour.

**Decision letter after peer review:**

Thank you for submitting your article "Reversal of contractility as a signature of self-organization in cytoskeletal bundles" for consideration by *eLife*. Your article has been reviewed by two peer reviewers, one of whom is a member of our Board of Reviewing Editors, and the evaluation has been overseen by Anna Akhmanova as the Senior Editor. The following individual involved in review of your submission has agreed to reveal their identity: Francois Nedelec (Reviewer #2).

The reviewers have discussed the reviews with one another and the Reviewing Editor has drafted this decision to help you prepare a revised submission.

Summary:

The author examines the dynamics of a circular bundle of filaments predicting regimes in which the bundle can contract or expand, in the presence of passive crosslinkers and crosslinking motors. The theory is done in 1D, following a mean-field approach. While all the assumptions are fairly standard in the field, their combination is interesting. Particularly filament treadmilling (assembly at the barbed end that is exactly compensated by disassembly at the pointed end) effectively provides a 'motor' quality to the crosslinkers, as they appear to move towards the pointed end, while the true motors move in the opposite direction. As a consequence, the system can be tuned to be contractile or extensile by changing the treadmilling velocity, which affects the speed of motors and crosslinkers move toward the ends. Treadmilling also leads to filament movements, but within the assumed of periodic boundaries, this has little impact on the overall length of the bundle. The buckling of filaments is not considered. The results are not entirely surprising, but the possibility for a network to expand is insufficiently discussed in the literature, although it is certainly an important behavior of cytoskeletal systems. The theory itself is clearly explained and convincing. It is of high quality and in line with that the author and others have done before on contraction, but now including extension, and this is new.

Minor points:

"Our prediction crucially rests on a simultaneous theoretical treatment of the filament, motor and crosslink dynamics detailed in the subsection “Motor and crosslink dynamics”. Previous studies of contractility mechanisms only involved partial treatments, whereby the time scale associated with one of these dynamics was effectively assumed to be much larger or smaller than the others."

We would agree that for most previous work this is true, but in Belmonte, Leptin and Nédélec, 2017, crosslinkers and motors were treated symmetrically, without making an assumption relating to different time scales. Please, note the supplementary material of this work, is not a simulation, but an analytical result.

Figure 2 legend: We were confused at first that ξ and thus *ρ*_m_(ξ) would be measured with L=0 at the barbed end, and not the other way around, which seems more natural (0 = minus-end).

"In the opposite limit where crosslinks are absent, motors slide filaments along at their unloaded velocity without producing any stresses, which rules out both contraction and extension [Lenz et al., 2012a]"

This is only true if one neglects the viscous drag, and this assumption should be repeated here ('without any stress' is an exaggeration).

"Indeed, while extension has been observed in numerical simulations reported in Belmonte, Leptin and Nédélec, 2017, this behavior was directly tied to a choice of somewhat exotic motor end-dwelling properties that favor anti-sarcomeric structures."

Belmonte, Leptin and Nédélec, 2017, discussed the consequence of a 'depletion zone' (called with a different name: antenna effect) on which is a cornerstone of the current work, and have predicted that they would lead to the extension of the network. We fail to see how this can be reduced to 'anti-sarcomeric structures' unless this would apply similarly to the current work.

"This system additionally does not manifest a transition from contraction to extension as the parameters of the model are continuously varied.". Please, refer to Figure 3D [Belmonte, Leptin and Nédélec, 2017] and correct (or delete) the statement.

"Another instance of bundle extension is reported in Kruse and Sekimoto, 2002. In that model, the extensile behavior stems from the interaction between parallel, not antiparallel filaments, which is at odds with experimental observations that tension generation in actomyosin requires antiparallel filaments [Reymann et al., 2012]." This statement appears inconsistent because it compares extension to contraction, and it seems possible to generate opposite forces, on different configurations (parallel vs. antiparallel filaments). At least that would be the expectation of many readers, and the statement, therefore, does not work. Perhaps the author could find something more positive to say about [Kruse and Sekimoto, 2002] which lead the way to the submitted work. We see it as a clear demonstration that the depletion zone can lead to movements between two parallel filaments, and thus break the symmetry that impairs extension/contraction in general.

Subsection “Quantitative Aspects and Alternative Models” seems flawed, as we expect the balance of forces during cytokinesis for echinoderms and *S. pombe* to be very different, one having a cell wall that dominates the mechanical behavior. Thus it is invalid to extrapolate from one condition to the other.

A number of formulations: "has been suggested to occur in some actin-myosin systems [Wollrab et al., 2018]", "somewhat exotic", "Finally, the motor properties leading to extension in this model may be relevant for microtubule systems, but are not for actin" suggest that the author does not believe that end-dwelling motors could exist for actin.

These statements are surprising especially in a theory paper. Can we really consider our knowledge of the cytoskeleton complete enough today to exclude a possibility that is already accepted for microtubules?

Finally, this comment is also relevant to the Discussion, specifically the fact that the absence of certain features can be taken as a proof of some other mechanism: I would argue that we do not know all the mechanisms operating in a cytoskeletal system, and with this in mind, can the author be confident enough to use the absence of a predicted behavior to make a strong conclusion about some other mechanism?

---

## [Author Response]

Minor points:"Our prediction crucially rests on a simultaneous theoretical treatment of the filament, motor and crosslink dynamics detailed in subsection “Motor and crosslink dynamics”. Previous studies of contractility mechanisms only involved partial treatments, whereby the time scale associated with one of these dynamics was effectively assumed to be much larger or smaller than the others."We would agree that for most previous work this is true, but in Belmonte, Leptin and Nédélec, 2017, crosslinkers and motors were treated symmetrically, without making an assumption relating to different time scales. Please, note the supplementary material of this work, is not a simulation, but an analytical result.

I thank the reviewers for critiquing this sentence. What I meant was that this work was to my knowledge the first to consider motor, crosslink as well filament treadmilling dynamics. Belmonte, Leptin and Nédélec, 2017 only describes the first two, and thus effectively assumes that treadmilling is much slower (it does consider filament turnover in one instance, but through a schematic filament removal/addition simulations protocol that is quite different from a detailed polymerization mechanism; additionally this protocol is not included by its analytical approach). I have clarified that the reference to filaments in this sentence regards their treadmilling/polymeration through the following modification of the sentence:

“Our prediction crucially rests on a simultaneous theoretical treatment of the filament polymerization, motor and crosslink dynamics detailed in subsection “Motor and crosslink dynamics”. […] We show in subsection “Self-organization and force distribution” that the coupled dynamics of these three elements induces a spatial organization of motors and crosslinks along the filaments”.

On the issue of numerical vs. analytical results, I have made the following addition to the conclusion highlighting the analytical contents of Belmonte, Leptin and Nédélec, 2017:

“[…] our analytical approach allows us to derive a complete phase diagram for self-organized motor-filament bundles. References Lenz et al., 2012b; Lenz,

2014; Belmonte, Leptin and Nédélec, 2017 derive similarly useful analytical results applicable to situations dominated by filament nonlinearity or motor dwelling.”

Figure 2 legend: We were confused at first that ξ and thus ρ_m_(ξ) would be measured with L=0 at the barbed end, and not the other way around, which seems more natural (0 = minus-end).

I added a comment stressing the convention used to minimize the risk of a reader being confused by this:

“Note that the coordinate ξ is measured from the filament’s plus end” (legend of Figure 2).

"In the opposite limit where crosslinks are absent, motors slide filaments along at their unloaded velocity without producing any stresses, which rules out both contraction and extension [Lenz et al., 2012a]"This is only true if one neglects the viscous drag, and this assumption should be repeated here ('without any stress' is an exaggeration).

This is correct. One has to keep in mind that viscous drag is negligible compared to the other forces discussed here, as currently discussed in the Discussion section. Following the reviewers’ recommendation I now explicitly state this assumption earlier in the section:

“Finally we neglect viscous drag, as further argued in the Discussion”.

"Indeed, while extension has been observed in numerical simulations reported in Belmonte, Leptin and Nédélec, 2017, this behavior was directly tied to a choice of somewhat exotic motor end-dwelling properties that favor anti-sarcomeric structures."Belmonte, Leptin and Nédélec, 2017 discussed the consequence of a 'depletion zone' (called with a different name: antenna effect) on which is a cornerstone of the current work, and have predicted that they would lead to the extension of the network. We fail to see how this can be reduced to 'anti-sarcomeric structures' unless this would apply similarly to the current work.

I take the reviewers’ overall point that my previous discussion of Belmonte on was needlessly critical, and have modified the language to offer a more nuanced discussion.

On the specific issue of the antenna effect, I now specifically mention its presence in the work of Belmonte, keeping in mind that it is only qualitatively discussed in the Discussion section rather than quantitatively modeled as I do here. I removed the “anti-sarcomeric” comment:

“Extension has indeed been observed in numerical simulations reported in Belmonte, Leptin and Nédélec, 2017, but only in situations where the motors were assumed to dwell at the filament ends. The Discussion section of this reference does however qualitatively predict that extension should result from an “antenna” mechanism similar to the depletion quantitatively modeled here.”

"This system additionally does not manifest a transition from contraction to extension as the parameters of the model are continuously varied.". Please, refer to Figure 3D [Belmonte, Leptin and Nédélec, 2017] and correct (or delete) the statement.

The original idea behind this comment was to contrast a situation where the transition is induced by a change in the composition of the system vs. its intrinsic parameters. This is however a very inessential point, and I deleted it according to the reviewer’s suggestion.

"Another instance of bundle extension is reported in Kruse and Sekimoto, 2002. In that model, the extensile behavior stems from the interaction between parallel, not antiparallel filaments, which is at odds with experimental observations that tension generation in actomyosin requires antiparallel filaments [Reymann et al., 2012]." This statement appears inconsistent because it compares extension to contraction, and it seems possible to generate opposite forces, on different configurations (parallel vs. antiparallel filaments). At least that would be the expectation of many readers, and the statement, therefore, does not work. Perhaps the author could find something more positive to say about Kruse and Sekimoto, 2002, which lead the way to the submitted work. We see it as a clear demonstration that the depletion zone can lead to movements between two parallel filaments, and thus break the symmetry that impairs extension/contraction in general.

The reviewers are correct in pointing out that my statement is confusing due to its clumsy use of the word “tension”, which may be taken as a synonym for “contraction”. I modified it slightly to more accurately portray the underlying reasoning:

“[…] which is at odds with experimental observations that bundle-wide force generation in actomyosin requires antiparallel filaments [Reymann et al., 2012].”

I do not agree with the characterization of Kruse and Sekimoto, 2002, as a proof that depletion zones (in the sense of the present manuscript, which is close to the “antenna effect” of Belmonte, Leptin and Nédélec, 2017) can lead to movements between parallel filaments. Indeed Kruse and Sekimoto, 2002, is not concerned with the depletion/antenna effect, but rather with motor traffic jams, which essentially give rise to an emergent motor end-dwelling.

Subsection “Quantitative Aspects and Alternative Models” seems flawed, as we expect the balance of forces during cytokinesis for echinoderms and *S. pombe* to be very different, one having a cell wall that dominates the mechanical behavior. Thus it is invalid to extrapolate from one condition to the other.

I agree with the reviewers that the extrapolation discussed in the first paragraph of the subsection “Quantitative aspects and alternative models” (and only there) is a bit of a stretch. I have thus replaced the numerical force extrapolated from echinoderms by a direct fission yeast measurement, which gives a result consistent with the one produced by the model using fission yeast parameters:

“The magnitude of the forces and velocities predicted by our model are on par with those found in vivo, e.g., in the cytokinetic ring of fission yeast. Indeed, setting L = 1.4µm, […] we find a contractile force T ' 120pN comparable with the ring tension of 390pN measured in fission yeast protoplasts (Stachowiak et al., 2014).”

A number of formulations: "has been suggested to occur in some actin-myosin systems [Wollrab et al., 2018]", "somewhat exotic", "Finally, the motor properties leading to extension in this model may be relevant for microtubule systems, but are not for actin" suggest that the author does not believe that end-dwelling motors could exist for actin.These statements are surprising especially in a theory paper. Can we really consider our knowledge of the cytoskeleton complete enough today to exclude a possibility that is already accepted for microtubules?

My position is not to exclude this possibility, but to point out that end dwelling is certainly more speculative in actomyosin than in microtubules, and should thus not be our default assumption. It seems to me that this position is indistinguishable from that expressed in the reviewers’ own work [Belmonte, Leptin and Nédélec, 2017], where the Figure 2 presents non-dwelling motors as “actin-like” whereas Figure 3 presents a mixture of dwelling and non-dwelling as “microtubule-like”. To better reflect this in the text I have modified the formulations that the reviewers objected to.

In the order of the citations above: “and could occur in some actin-myosin systems [Wollrab et al., 2018]”

“in situations where the motors were assumed to dwell at the filament ends”

[sentence deleted] (subsection “Qualitative predictions”).

Finally, this comment is also relevant to the Discussion, specifically the fact that the absence of certain features can be taken as a proof of some other mechanism: I would argue that we do not know all the mechanisms operating in a cytoskeletal system, and with this in mind, can the author be confident enough to use the absence of a predicted behavior to make a strong conclusion about some other mechanism?

I agree with the reviewer’s careful take on the issue, and have modified some of the language of the Discussion accordingly, replacing my previous uses of “ruling out” self-assembly mechanisms by “arguing against” and “tending to disqualify”.